# In Vitro Polyploidization of *Brassolaeliocattleya* Hybrid Orchid

**DOI:** 10.3390/plants12020281

**Published:** 2023-01-07

**Authors:** Joe Abdul Vilcherrez-Atoche, Jéssica Coutinho Silva, Wellington Ronildo Clarindo, Mateus Mondin, Jean Carlos Cardoso

**Affiliations:** 1Laboratory of Plant Physiology and Tissue Culture, Depart of Biotechnology, Plant and Animal Production, Center of Agricultural Sciences, Federal University of São Carlos (CCA/UFSCar), Rodovia Anhanguera, km 174, CP 153, Araras 13600-970, SP, Brazil; 2Master Science Graduate Program of Plant Production and Associated Bioprocesses, Center of Agricultural Sciences, Federal University of São Carlos, Araras 13600-970, SP, Brazil; 3Laboratório de Citogenética e Citometria, Departamento de Biologia Geral, Centro de Ciências Biológicas e da Saúde, Universidade Federal de Viçosa, Viçosa 36570-000, MG, Brazil; 4Departamento de Genética, Escola Superior de Agricultura “Luiz de Queiroz”, Universidade de São Paulo, Piracicaba 13400-970, SP, Brazil

**Keywords:** orchid hybridization, in vitro culture, seedling, colchicine, polyploids, flow cytometry

## Abstract

The *Cattleya* (Orchidaceae–*Laeliinae* subtribe) intergeneric hybrids, such as *Brassolaeliocattleya* (*Blc*.), have great ornamental value, due to their compact-size, with large and high color diversity of flowers. Artificial induction of polyploidy brings agronomic, ornamental and genetic benefits to plants. Polyploidization efficiency depends on factors, such as the type of antimitotic, polyploidization method, concentrations, exposure times and type of explant. This study aimed to develop a protocol to polyploidize *Blc*. orchids, by testing two types of explants (seeds and protocorms), concentrations and exposure times to colchicine. The effects of colchicine on the in vitro development of explants were also investigated. The responses of explants to colchicine depended on the concentrations, exposure time and the interaction of these factors. Flow cytometric analysis evidenced high endopolyploidy and allowed the separation of polyploidized (4C, 8C and 16C peaks) from non-polyploidized (only 2C and 4C peaks) plants. The highest percentage of polyploid plants was regenerated from protocorms (16.4%) treated with colchicine instead of seeds (3.2%). Protocorms treated with colchicine at 500–750 μM for 18 h resulted in the best percentage of polyploidization. Additionally, in vitro natural polyploidization using protocorms was reported (11.5%). Cytological analyses allowed an estimation of the number of chromosomes of the parents (≡70), polyploidized (≡140) and non-polyploidized progeny (≡70).

## 1. Introduction

Orchids of the genus *Cattleya* (Orchidaceae) are commonly called the “queens of orchids” [1], whose flowers are characterized by their large and wide petals in relation to the more elongated sepals, and have a lip with a great ornamental value [2]. In *Cattleya*, the currently commercialized plants are hybrids, obtained by interspecific or intergeneric hybridization, which has been the most widely breeding technique used to obtain a great diversity of commercial hybrids [3].

*Cattleya* hybrids show an excellent ornamental quality and acceptance in the international potted flower market [4] but the long juvenile period, the highly genotype-specific response to flowering and the short lifespan of flowers hampered the efficient development of large-scale production [3,5] similar to developed with other orchid genera, such as *Phalaenopsis, Dendrobium* and *Oncidium* hybrids, with greater commercial importance.

In orchid breeding programs, the addition of biotechnological tools to conventional hybridization, such as in vitro polyploidization, could be used to obtain superior genotypes with desirable characteristics that have not yet been obtained by conventional breeding programs [6].

Polyploidy is a phenomenon associated with organisms with more than two sets of chromosomes in their cells [7]. Polyploidy in plants can be naturally achieved either by endoreduplication [8] or by the fusion of unreduced gametes [9] but has widely been achieved by artificial induction using chemicals with antimitotic action. In plants, the most common antimitotic used is colchicine [10,11].

Colchicine inhibits microtubule formation in the chromatic spindle causing nondisjunction and subsequent duplication of chromosomes within plant cells, generating polyploid cells [12], which develop and regenerate into polyploid plants. Polyploidization produces interesting changes and new features in cultivated plants, ranging from morphological, horticultural and cellular changes that are affected by physiological, biochemical and genetic characteristics, with some improvements in relation to diploid organisms [12]. For example, polyploidization in flowers results in changes of development and architecture of plants and also in floral structures, as well as in the flowering season and number of flowers [13]. Polyploidization has been used in breeding programs in different ornamental plants, such as *Salvia coccinea* cv. Coral Nymph, *Gladiolus grandiflorus* and *Chrysanthemum carinatum*, in which polyploid plants showed larger flowers, with thicker petals and a longer shelf-life [14,15,16]. Additionally, in the orchid industry and trade, production of new polyploid cultivars usually results in superior ornamental characteristics compared to diploid cultivars [17].

However, the efficiency to induce chromosome duplication under in vitro conditions requires the development of a protocol including the methodology containing all the steps of in vitro cultivation, as well as the methods used for treating plant tissues with antimitotic chemicals. The factors affecting most the efficiency of polyploidization include the genotype, type of explant, culture medium and cultivation conditions, and those related to the antimitotic agent, such as molecule used and induction method, concentration and exposure time, and the method used to confirm polyploidization [10].

Different protocols for artificial induction of polyploid plants have already been developed for the main commercial genera of orchids, such as *Phalaenopsis* [18], *Cymbidium* [19], *Dendrobium* [20] and *Oncidium* [21]. In *Cattleya*, there are only two studies on the induction of polyploidy using only species, such as *Cattleya tigrina* [22] and *C. intermedia* [23]. In addition, there is limited knowledge about ploidy levels of interspecific and intergeneric hybrids as a result of hybridization, and efficient protocols aimed to achieve the polyploidization of commercial *Cattleya* hybrids, such as *Brassolaeliocattleya*. In this context, the objective of this study was to test different types of explants subjected to different exposure times and colchicine concentrations, aiming to develop a protocol for polyploidization and in vitro artificial chromosome duplication of a *Brassolaeliocattleya* hybrid. In addition, we also studied the different effects of colchicine on the in vitro development and growth of explants treated with this alkaloid.

## 2. Results

### 2.1. Colchicine Effects on Explant Development

(a)Seeds

The percentage of seeds containing embryos (total five repetitions of 100 seeds counted under microscopy) of the crossing between *Blc.* Haw Yuan Beauty × *Blc.* Goldenzelle “LC” was 81.25% [(number of seeds containing embryos/total number of seeds) * 100)].

Increases in colchicine concentration or exposure time caused a decrease in the percentage of germinated seeds (Table 1). The percentage of seeds that developed into seedlings, instead of protocorms, only increased using the highest concentration of 1000 μM (Table 1). On the contrary, increasing exposure time led to increases in protocorm development (54.99% to 68.36%), instead of seedlings with roots and shoots, which decreased from 45.01% to 31.64% (Table 1).

Both increases in colchicine concentration and exposure time had an effect on the total fresh weight, with a reduction of 2.75 g at 1000 μM colchicine (11.77 g/vial) in relation to the control treatment (14.52 g/vial) (Table 1).

(b)Protocorms

Increasing concentrations of colchicine resulted in a gradual decrease of protocorm survival (Table 2). The type of regeneration of protocorms, either by the proliferation of protocorm like-bodies (PLBs) or by the formation of new plantlets, were not affected by colchicine concentration (Table 2). In addition, increases in colchicine concentration gradually reduced the total fresh weight of tissues regenerated from protocorms, from 12.75 g (control) to 8.66 g (at 1000 μM), a reduction of 32% total fresh weight (Table 2).

Exposure time to colchicine had no effects on explant survival (Table 2). However, the best percentage of regeneration via plantlet formation, instead of PLBs proliferation, was reported at 12 h time exposure (Table 2). The largest difference occurred when treated for 6 h compared to longer exposure times, in which there was a reduction in the proliferation of PLBs and an increase in protocorms regenerated into plantlets (Table 2).

Colchicine also had important physiological effects on protocorms, resulting in the death of part of the protocorms and tissues exposed to colchicine. This was observed by the difference in color in part of the explants treated with the alkaloid, which showed a brown color, demonstrating partial or total death of the treated tissues (Figure 1A), compared to untreated protocorms (Figure 1B).

Although most protocorms showed a brown-color, after treatment with colchicine, the emergence of new points of regeneration of PLBs in tissues were observed, demonstrating that phytotoxic effects of colchicine did not completely affect the tissue, thus allowing their regeneration (Figure 2).

### 2.2. Flow Cytometry and Cytogenetic Analysis

Specific and high-quality peaks (2C, 4C, 8C or 16C) were obtained for the analyzed individuals (Figure 3). Flow cytometry confirmed the hybrid origin of the obtained seedlings (Figure 4). Nuclear 2C value showed that the female parent (*Blc*. Haw Yuan Beauty) has a higher DNA content (2C = 7.79 pg) than the male parent (*Blc*. Goldenzelle ‘LC’, 2C = 5.79 pg) (Figure 3A,B). All the progenies from the control (without colchicine treatment) showed intermediary peaks (Figure 3C and Figure 4) and nuclear 2C value (2C = 6.86 pg). Flow cytometry also confirmed the polyploidized plants, which showed higher nuclear DNA content (2C = 14.37 pg) (Figure 3D).

Flow cytometry histograms showed the presence of cells with different DNA ploidy levels, e.g., 2C, 4C, 8C and 16C cells, demonstrating that the leaves of progeny showed a high level of endopolyploidy (Figure 3C,D). Thus, the distinction between polyploid and non-polyploid plants was based on the following parameters: diploid plants were considered those showing only cells in 2C, 4C and sometimes 8C channels (Figure 3C), while polyploid plants showed cells only in 4C, 8C and 16C channels (Figure 3D).

In general, polyploidized plants presented different morphology and architecture compared to non-polyploidized plants (Figure 5). Polyploid plants had oblong-shaped leaves with greater thickness, width and intensity of green color (Figure 5A), unlike non-polyploid plants, which showed lanceolate-shaped leaves with lower width, thickness and a light green color (Figure 5B).

Chromosome counting of the parents showed that the estimated number of chromosomes was approximately 70 (Figure 6B), which is similar to the F1 progenies (≡70) (Figure 6A). Root tips of polyploid plants, analyzed by flow cytometry, confirmed the occurrence of in vitro polyploidization (≡140 chromosomes), especially in treatments with colchicine (Figure 6C,D).

### 2.3. Protocorms Resulted in the Best Polyploidization Rates Compared to Seeds

Only five out of 154 progenies (3.21%) derived from seeds were polyploidized using colchicine (Figure 7A). Polyploid plants were derived only from seeds treated at 500, 750 and 1000 μM (Figure 7A). Differently and more effective than seeds, the use of protocorms treated with colchicine (Figure 7) resulted in a total of 35 polyploid plants out of the total of 213 analyzed (16.43% polyploidization efficiency). The best frequencies of polyploidization (44 and 46%) were obtained using 1000 μM colchicine for 18 h and 500 μM colchicine for 6 h). Positive correlation between colchicine concentration and frequency of polyploidized seedlings was observed for colchicine treatment of protocorms (Figure 7B). The treatment time was also positively correlated with frequency of polyploidization until 18 h. The treatment for 24 h resulted in a reduction in polyploid frequency (14.6%) and was lower compared to shorter induction times (Figure 7C). In addition, natural polyploidization occurred in low frequency (11.5%) in water-treated protocorms (Figure 7A).

## 3. Discussion

### 3.1. Colchicine Switched the In Vitro Development of Blc. Orchid

Colchicine has been the main substance used for chromosome duplication in plants. However, colchicine phytotoxicity was reported in different types of tissues used as explants, where the different levels of toxicity depend on the concentrations and exposure times used to treat the plant material [24].

In orchids, such as *Bletilla striata*, the treatment with the highest concentration of colchicine and longer exposure time (0.4% colchicine for 9 days) resulted in the lowest seed germination rate [25]. Chung et al. [13] also induced polyploidy using hybrid seeds of *Calanthe* (*C. discolor* × *C. sieboldii*) (Orchidaceae) and reported a slight gradual decrease in seed germination as a consequence of increasing concentration and exposure time of seeds to colchicine or oryzalin.

Lone et al. [26] observed a reduction in the survival rate of protocorms of *Cattleya tigrina* at the highest concentration of colchicine (1%), showing survival rates of 100%, 96% and 84%, with 24, 48 and 72 h of exposure, respectively, and compared to the control (100%). Similar results were reported for *Vanda* (Orchidaceae) protocorms treated with colchicine, aiming to induce polyploidy. The tissue response to colchicine resulted in greater accumulation of phenolic compounds, resulting in limited development of treated explants [27].

In the current experiment, lower survival percentages (between 40 and 80%) were observed after treating with colchicine. Reduction in growth of plant tissues treated with colchicine has been attributed to its negative effects on meristematic cells, producing abnormal cycles during cell division [28] and high concentrations of colchicine cause a decrease in mitotic division rates in treated tissues [29], which cause decreases in plant growth. These negative effects of colchicine were also observed in the present study with the hybrid progeny from *Blc.* Haw Yuan Beauty x *Blc.* Goldenzelle “LC”.

### 3.2. Colchicine Is Efficient at Inducing Polyploidization in Blc. Orchid

Chung et al. [13], using seeds of *Calanthe* (*C. discolor* × *C. sieboldii*) as explants, reported an increase in the percentage of tetraploids as a function of time of treatment (from 3 to 7 days) and concentration (0.05% to 0.1%) of colchicine. *Calanthe* hybrid seeds treated with 0.1% colchicine for 3 and 7 days produced a high percentage of polyploids, 74% and 81%, respectively [13].

Among the differences between the present study with *Blc.*, using seeds as explants, and that conducted by Chung et al. [13], was the time of exposure of seeds to colchicine and the genotype. While the latter authors adopted 3–7 days, the present study with *Blc.* used 6–24 h (or 1 day). Despite this, in *Blc*. it was observed that the longest time of treatment with colchicine (24 h) was not the one with the highest frequency of polyploidized plants. Chung et al. [13] also concluded that the high percentage of polyploids in *Calanthe* is also due to its genome that allow the easy duplication of the number of chromosomes.

Even with a low rate of colchicine-treated seeds that developed into polyploidized plants in the present study on *Blc*., the small seed size enabled the treatment of a high number of seeds using a reduced amount of colchicine solution, a reagent with high cost and risks associated with its manipulation [6].

Different from seeds used as explants, increases in the percentage of polyploids in colchicine-treated protocorms of *Blc.* are correlated with colchicine concentrations and exposure times and were superior or similar (38–46%) to those previously obtained with other orchid species and hybrids. For example, in protocorms of the hybrid *Cymbidium sinenthese* ‘Lv mosu’ × *Cym. hybridum* ‘Shijieheping’, the treatment with 0.03% colchicine for 72 h generated the best result, with 36% polyploids [30]. PLBs of *Dendrobium chrysotoxum* treated with 0.04% colchicine for 24 h produced a polyploid frequency of 47% [31]. Additionally, protocorms of *Phalaenopsis equestris*, *Phal. fasciata* and *Phal*. Betty Hausermann showed high-frequency of polyploidization, 46%, using the treatment with 50 mg L^−1^ colchicine for 10 h [32].

Other interesting observation of our study are the presence of polyploidized seedlings from protocorms not treated with colchicine. Three main factors could explain such results: polyploid seedlings were generated during or as result of hybridization; the occurrence of natural endoreduplication in protocorm cells, which formed polyploidized PLBs and plantlets; the use of liquid medium and the presence of anaerobiosis combined with the time of treatment of the protocorms induces polyploidy.

Although polyploidization in response to hybridization has already been reported in orchids [33], this is not the case of natural polyploid plants from *Brassolaeliocattleya*, since flow cytometry of non-treated progenies showed intermediary nuclear DNA content between two parents. The hypothesis of endopolyploid cells inside protocorms that resulted in natural polyploid plants is strengthened by flow cytometry analysis that showed a high number of endopolyploid cells in tissues of protocorms of the studied hybrid.

Thus, the most accepted hypothesis is that the sections, manipulation and exposure of endopolyploid cells of the isolated protocorms resulted in the production of polyploid PLBs in the absence of colchicine. This result and conclusion were confirmed by Chen et al. [34], in which the sectioning of PLBs from *Phalaenopsis aphrodite* resulted in up to 34% polyploid PLBs. These authors [34] concluded that the main cause of polyploids regenerated from PLBs was derived from endopolyploid cells observed in their tissues. Furthermore, *Cattleya* is an orchid genus in which it has been observed endopolyploidy events in different tissues of several species, such as *C. trigina* [22], *C. trianae, C. grandis, C. guttata, C. labiata, C. cernua, C. tenius, C. elongata, C. crispata, C rupestres, C. aclandiae, C. amethystoglossa, C. pfisterii, C. rupestris, C. sincorana, C. loddigesii* and *C. granulosa* [35].

This spontaneous system for polyploidization of orchids using protocorms is extremely interesting from a practical point of view, as it does not require complex and additional procedures for using colchicine [18], a high-cost product that can pose health risks to the operator, whether in the preparation of solutions or in the treatments applied to plant tissues.

### 3.3. Ploidy Levels and Chromosome Counting in Blc. Orchids

Direct and indirect methods can be used for identification and confirmation of polyploid plants compared to diploid ones [36]. Flow cytometry is also considered a high-throughput system for early screening of polyploid plants in orchids [37].

The presence of endopolyploid cells, as reported for *Blc.* progenies in the present study, was also observed in other orchids, such as *Cattleya tigrina,* with ploidy levels of 2C and 4C and self-polyploidized seedlings resulting in cells with ploidy levels of 4C and 8C [22]. In our study, polyploidized plants showed only 4C and 8C peaks and at much lower frequencies, 16C, instead of 2C and 4C observed in most of non-colchicine treated protocorms.

As a consequence, seedlings of *Blc*. also showed changes in morphological features, and similar to observed in the leaves, such as thickness, length and intensity of green color in polyploids of *C. tigrina* [22], *Dendrobium formosum* [38], *Cymbidum lowianum* [39] *Phalaenopsis amabilis* and *Phal. amboinensis* [40]. The intensity of leaf color is used as a morphological marker for the identification of polyploidized plants [36], where it is believed that this effect is related to chromosomal duplication, causing an increase in the content of pigments and the enzyme production in polyploid plant cells [22].

Root tips of polyploid plants were analyzed by chromosome counting and confirmed the occurrence of in vitro polyploidization (≡140 chromosomes).

The combined results of cytogenetics with flow cytometry analysis confirmed that polyploidized plants contained twice the number of chromosomes and more than twice the DNA content of non-polyploidized plants. However, it was not possible to determine the exact number of chromosomes from the parents and progeny obtained from the cross between the two cultivars of *Brassolaeliocattleya* (*Blc*. Haw Yuan Beauty x *Blc*. Goldenzelle “LC”) used in the present study, which have ornamental and horticultural characteristics from *C. briegeri*, *C. intermedia*, *C. forbesi*, *C. loddigesii*, *C. dowiana*, *C. trianae*, *C. tenebrosa* and *C. bicolor* [41,42]. Cytological studies have shown that most *Cattleya* species mentioned above have 40 chromosomes [43,44], but there are also intraspecific chromosomal variations, such as in *C. trianae* and *C. bicolor* with 42 chromosomes [44,45]. Similarly, *Brassavola* and *Laelia* also belong to the subtribe *Laeliinae*, therefore they have 40 somatic chromosomes, in addition to some chromosomal variations of 42, 44 and 60 identified in the genus *Laelia* [45].

Molecular cytogenetic techniques allowed increasing the amount of information on the evolution of the karyotype in the subtribe Laeliinae [43,46]. A study on the evolutionary karyotype diversity in the subtribe *Laeliinae*, using molecular cytogenetics together with chromosome band analysis, demonstrated that *C. trianae* showed a fusion of a chromosome pair as a rearrangement mechanism. *Laelia gouldiana* presented a polyploid karyotype and *L. marginata* had a supernumerary chromosome [43] indicating the high degree of chromosomal variations common to this subtribe, to which the hybrid used in the present study belongs.

Difficulties encountered in obtaining the exact number of chromosomes in *Blc*. Haw Yuan Beauty × *Blc*. Goldenzelle “LC” were related to the high number of chromosomes found, in both parents and progeny, and the difficulty in finding perfect metaphases, as well as the low rate of cell divisions observed in root meristems during the in vitro culture of *Blc*. Haw Yuan Beauty x *Blc*. Goldenzelle “LC”.

## 4. Material and Methods

### 4.1. Plant Material, In Vitro Establishment and Growth

Seeds from a mature capsule, eight months old, were obtained from the crossing between two hybrid cultivars: *Brassolaeliocattleya* ‘Haw Yuan Beauty’ and *Blc.* Goldenzelle “LC” (Orchidaceae) germplasm collection at UFSCar, Araras, Brazil. Seeds were dried at room temperature for 24 h before storage in 1.5 mL eppendorf type vials at 8 °C.

Disinfection and in vitro seeding were carried out using a solution containing one volume of bleach (2.0–2.5% active chlorine) and nine volumes of autoclaved deionized water. Seeds were immersed in this solution for 12 min under agitation, followed by three washes in autoclaved deionized water. Seeding was performed in 30 mL Murashige and Skoog culture medium [47] with the macronutrient concentration reduced by half, with 2% sucrose (Synth^®^, Diadema-SP, Brazil), 1.2 g L^−1^ activated charcoal (Synth^®^), 0.1 g L^−1^ inositol (Synth^®^) and pH adjusted to 5.7 before the addition of 6.4 g L^−1^ agar (Agargel^®^, João Pessoa-PB, Brazil) inside glass flasks (240 mL capacity) covered with polypropylene caps. The culture media contained in the flasks was sterilized by autoclaving for 25 min at 121 °C and 1 atm.

### 4.2. Colchicine Treatment Procedures

Colchicine (Sigma-Aldrich^®^, Saint Louis, MO, USA) was prepared from a stock solution of colchicine, previously dissolved in a 1% (*v/v*) solution of dimethylsulfoxide (Synth^®^). Five concentrations were tested in this experiment: 0.00 μM; 250 μM; 500 μM; 750 μM and 1000 μM, combined with four immersion times: 6, 12, 18 and 24 h.

Treatments with colchicine were applied to seeds collected and stored from the *Blc*. hybrid and used as explants, and to protocorms also obtained from the germination of the same seeds, but after 90 days of in vitro cultivation in the culture medium described in Section 4.1. For seed treatment, 5 mg fresh mass of seeds per treatment and 15 protocorms per replication (60 per treatment) were used for this study (Figure 8). The graphical abstract with step-by-step experiment is also presented (Figure 9).

For the seeds, the different concentrations of colchicine were applied as pre-treatment, by the immersion of seeds in solutions containing different concentrations of colchicine, using a 250 mL Erlenmeyer flask kept in a horizontal rotary shaker at 60 rpm and in dark conditions at 25 ± 1 °C during exposure times. After exposure time, seeds were subjected to asepsis and inoculated in vitro in a germination culture medium, as described in Section 4.1.

In vitro protocorms, obtained after 90 days of seed germination, were selected and subjected to different concentrations of colchicine and exposure times. The colchicine solution was filter-sterilized using a Millipore Millex™ EMD syringe filter (<0.22 µm) and poured into a 250 mL Erlenmeyer flask containing liquid MS medium (with no activated charcoal and agar). After preparing the different concentrations of colchicine, protocorms were immersed and maintained in horizontal rotary shakers at 60 rpm in dark conditions at 25 ± 1 °C during the exposure times.

For each explant, controls were made for each exposure time, with all detailed procedures maintained, except for the addition of colchicine in contact with seeds or protocorms. At the end of the colchicine exposure times, all explants were washed with sterile deionized water three times to remove colchicine from the explants, and then cultured again in MS culture medium containing charcoal and agar, as described in Section 4.1. There was no addition of plant growth regulators to the culture medium.

Cultivation of seeds or protocorms was carried out at 25 ± 2 °C with lighting provided by Light-Emitting Diodes (LEDs) in the red and blue wavelengths (3:1), with a Photosynthetically Photon Flux Density (PPFD) of ≡ 50 µmol cm^−2^ s^−1^ and photoperiod of 16 h.

For both explants, a 5 × 4 factorial completely randomized design was adopted, with five concentrations of colchicine and four exposure times. In total, four replications were used per treatment, consisting of 240 mL glass flasks containing 30 mL culture medium with the seeds (at least 5.0 mg/vial) or protocorms (15/vial).

### 4.3. Effects of Colchicine on Seeds and Protocorms

Evaluations of in vitro germination, regeneration and development were specific to each type of explant:

(a)Seeds: Using the counting method, the percentage of seeds was evaluated by the presence or absence of the embryo inside the testa with the aid of an optical microscope (Nikon Eclipse e200, Nikon Instruments, Japan) with a 4X objective lens (Figure 1B). This included the percentage of germination 180 days after seeding; the percentage of embryos that developed into protocorms and/or seedlings based on fresh mass calculation of each type of development after germination; the total fresh weight of plant tissue obtained per vial.(b)Protocorms: After 180 days of culture following the treatment with colchicine, the percentages of survival and death of protocorms, the percentages of PLBs proliferation and the regeneration into plantlets and the total fresh weight obtained per vial were recorded.

### 4.4. Flow Cytometry Analysis

The nuclear 2C value was measured from eight seedlings (3–4 months old) of each treatment of seeds, and from 13 seedlings (7–8 months age) of each treatment of the protocorms in MS culture medium. The internal standard used for the analysis was *Solanum lycopersicum* L., 1753, ‘Stupické’ (2C = 2.00 pg) [25].

A leaf fragment of ~2 cm^2^ from individuals of *Blc.* Haw Yuan Beauty × *Blc.* Goldenzelle “LC” and the internal standard were simultaneously chopped [48] for about 30 sec in a Petri dish containing 0.5 mL OTTO-I lysis buffer [49] supplemented with 50 µg mL^−1^ RNAse (Sigma^®^) and 2 mM dithiothreitol (Sigma^®^) [50] and incubated for 3 min. A total of 0.5 mL of the same buffer was added and the suspension was filtered through a 30 µm diameter nylon mesh (Partec^®^ Gmbh, Munster, Germany) in a 2.0 mL microtube. After centrifugation at 100× *g* for 5 min, the supernatant was discarded and 100 μL of the same buffer was added to the precipitate; the material was vortexed and incubated for 10 min.

Subsequently, 0.5 mL modified OTTO-II staining buffer [49,50] (400 mM Na_2_HPO_4_.H_2_O, 2 mM dithiothreitol (Sigma^®^), 50 µg mL^−1^ RNAse (Sigma^®^) and 75 µg mL^−1^ propidium iodide (PI (Sigma^®^), excitation/emission wavelengths: 480–575/550–740 nm) was added to the suspensions.

Suspensions were filtered through 20 µm nylon mesh (Partec^®^) into the reading tubes (Partec^®^) and kept for 30 min in the dark to stain the nuclei. Then, suspensions were analyzed in a flow cytometer (BD Accuri C6 flow cytometer, Accuri cytometers, Belgium) equipped with a 488 nm laser to promote PI excitation and PI emission to the FL2 (615–670 nm) and FL3 (>670 nm).

Fluorescence peaks of the G0/G1 nuclei of each individual of *Blc*. Haw Yuan Beauty × *Blc*. Goldenzelle “LC” and the internal standard were analyzed from histograms using BD Accuri™ C6 software. G0/G1 peaks with a coefficient of variation (CV) less than 5% were considered to determine the level of DNA ploidy. The nuclear 2C value of each individual, in picograms (pg), was calculated using the formula below:

DNA content of each individual (pg) = [(mean channel of peak G0/G1 of each individual of *Blc*. Haw Yuan Beauty × *Blc*. Goldenzelle “LC”) *2.00 pg *S. lycopersicum*]/(mean channel of peak G0/G1 from *S. lycopersicum*) *Blc.* Haw Yuan Beauty and *Blc.* Goldenzelle “LC” cultivars used as parents were also analyzed by flow cytometry.

### 4.5. Chromosome Counting

Root tips were collected from seedlings in in vitro conditions. Root tips (~2 cm) were pre-treated in a solution of 8-hydroxyquinoline (8-HQ) (Sigma^®^) at 300 ppm and cyclohexamine (Sigma^®^) at 25 ppm (19:1) for 24 h at a controlled temperature of 27 °C. The pre-treated roots were fixed in a 3:1 Carnoy solution for 24 h at a controlled room temperature of 27 °C.

The Feulgen method was used to stain the roots [51]. Roots were washed twice for five min with distilled water and hydrolyzed in a 5N HCl solution (Merck KGaA, Darmstadt, Germany) at 60 °C for 12 min. Afterwards, two washes were made with distilled water for five min and hydrolyzed roots were incubated in Schiff’s reagent for 45 min in the dark.

Stained roots were treated twice in a 0.01 M citrate buffer solution for 5 min each, followed by enzymatic digestion using a mixed solution of cellulase (Serva 16420, Germany, final concentration of 1.4 U mL^−1^) and pectinase (Calbiochem 515883, Germany, final concentration of 29.4 U mL^−1^) (1:1) for one hour of incubation.

After digestion, roots were placed in a citrate buffer solution on ice until mounting the slides. The protocol described by Mondin and Aguiar-Perecin [51] was followed for preparation of cytological slides containing mitotic metaphases, in which the root was placed in a 45% acetic acid solution (Merck KGaA, Darmstadt, Germany) for ~2 min and then the root meristem was macerated on a slide with a drop of 1% acetic carmine. A coverslip was placed on the macerated tissue and heated with a lamp for later crushing/squashing. For chromosome counting, slides were analyzed using a Zeiss Axiophot 2 microscope using the appropriate filter. Images were acquired by the PCO CCD camera and digitized in the IKARUS software (Metasystems, Germany).

Images of cells containing mitotic metaphases were captured with a 100X objective lens. Image J software was used to analyze and count the chromosomes.

### 4.6. Statistical Analysis

All data were analyzed using the AgroEstat Online software (http://www.agroestat.com.br/), in which homogeneity and homoscedasticity tests, and analysis of variance (ANOVA) were run, and when a difference was detected, the means were compared by Tukey’s test (*p* < 0.05 and <0.01). In addition, the frequency of polyploids from different concentrations and treatment time with colchicine were submitted to regression analysis and the coefficient of correlation (r values) were tested using Student’s *t*-test.

## 5. Conclusions

This study reports the different effects of colchicine on in vitro growth and development of seeds and protocorms of the *Brassolaeliocattleya* hybrid, from the crossing between *Blc*. Haw Yuan Beauty × *Blc*. Goldenzelle “LC”. In addition, we developed an efficient methodology for polyploidization of this orchid of high ornamental value. In addition, the potential of a colchicine-free polyploidization was demonstrated by using individualized protocorms as explants. Flow cytometry and cytological analysis were efficient in estimating ploidy and separating polyploid plants from non-polyploid plants.

## Figures and Tables

**Figure 1 plants-12-00281-f001:**
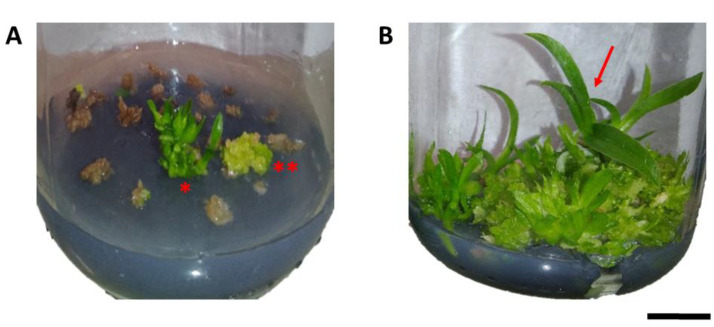
Effects of colchicine on in vitro protocorm survival of *Blc*. Haw Yuan Beauty × *Blc*. Goldenzelle “LC”: (**A**) protocorms cultured in MS medium for 120 days after treatment with colchicine at a concentration of 1000 μM and exposure time of 24 h, showing dead tissues, regeneration of PLBs ** and plantlets *; (**B**) control treatment with protocorms in plant regeneration (red arrow). Scale bar = 1 cm.

**Figure 2 plants-12-00281-f002:**
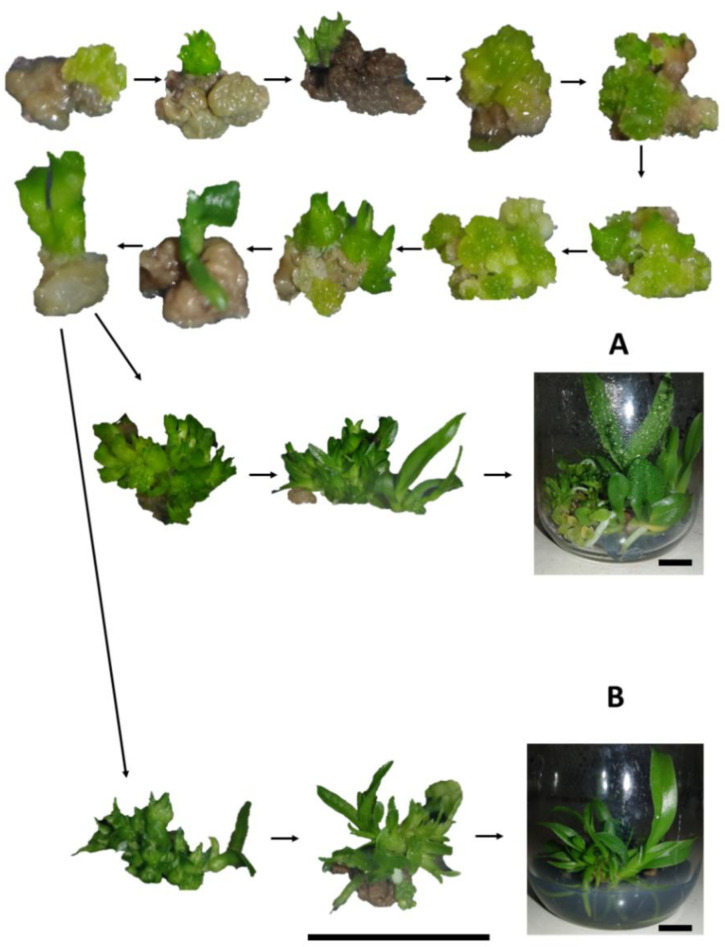
Regeneration of progeny of *Blc*. Haw Yuan Beauty × *Blc*. Goldenzelle “LC” in response to colchicine treatment of protocorms: (**A**) plantlets regenerated from protocorms treated with colchicine; (**B**) plantlets regenerated from protocorms not treated with colchicine. Scale bar = 1 cm.

**Figure 3 plants-12-00281-f003:**
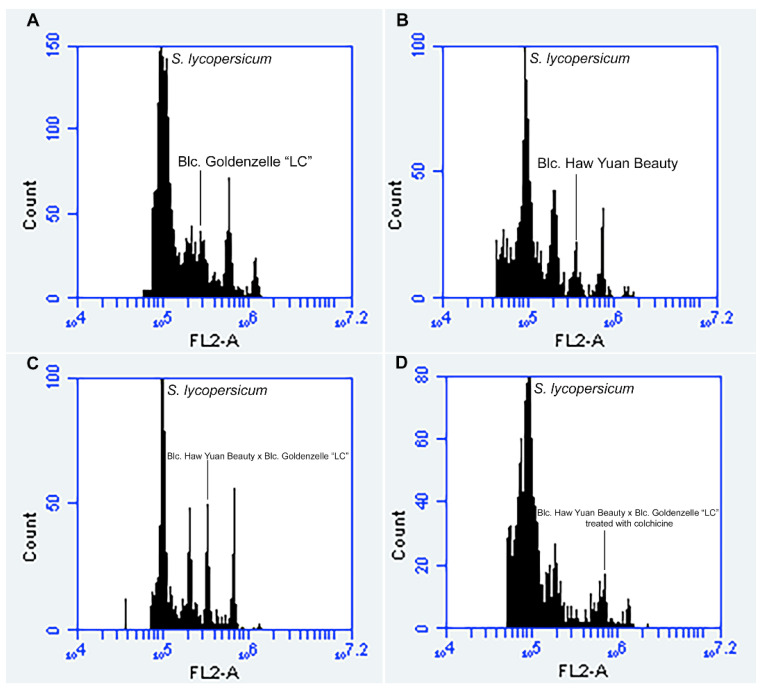
Flow cytometry histograms with nuclear 2C value measurements using the internal standard *S. lycopersicum* (2C = 2.00 pg): (**A**) male parent *Blc.* Goldenzelle “LC” with 2C = 5.79 pg; (**B**) female parent *Blc*. Haw Yuan Beauty with 2C = 7.79 pg; (**C**) *Blc*. Haw Yuan Beauty × *Blc*. Goldenzelle “LC” not treated with colchicine with 2C = 6.86 pg; (**D**) *Blc.* Haw Yuan Beauty × *Blc.* Goldenzelle “LC” treated with colchicine with 2C = 14.37 pg.

**Figure 4 plants-12-00281-f004:**
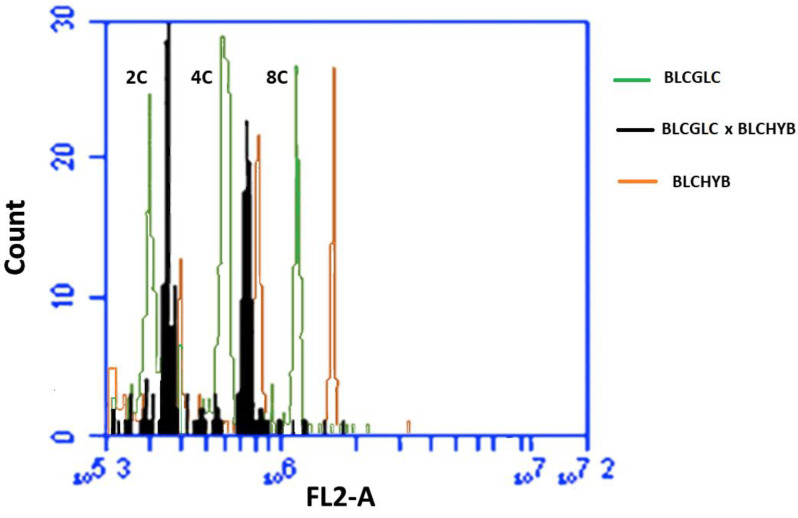
Comparison of the 2C, 4C and 8C peaks channels between *Blc.* Haw Yuan Beauty (BLCHYB) (orange line), *Blc.* Goldenzelle “LC” (BLCGLC) (green line) and the progeny of *Blc.* Haw Yuan Beauty × *Blc.* Goldenzelle “LC” not treated with colchicine (black line).

**Figure 5 plants-12-00281-f005:**
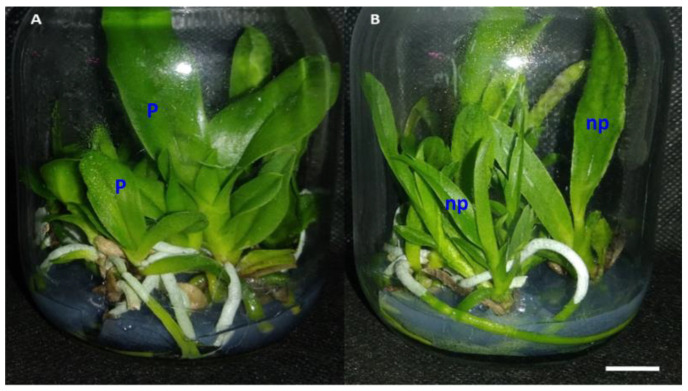
Development and in vitro rooting of plantlets regenerated from protocorm explants treated and not treated with colchicine: (**A**) vial containing only polyploid plants (P) of the progeny from *Blc*. Haw Yuan Beauty × *Blc.* Goldenzelle “LC”; (**B**) vial containing non-polyploidized plants (np) of the progeny from *Blc.* Haw Yuan Beauty × *Blc.* Goldenzelle “LC” and not treated with colchicine. Scale bar = 1 cm.

**Figure 6 plants-12-00281-f006:**
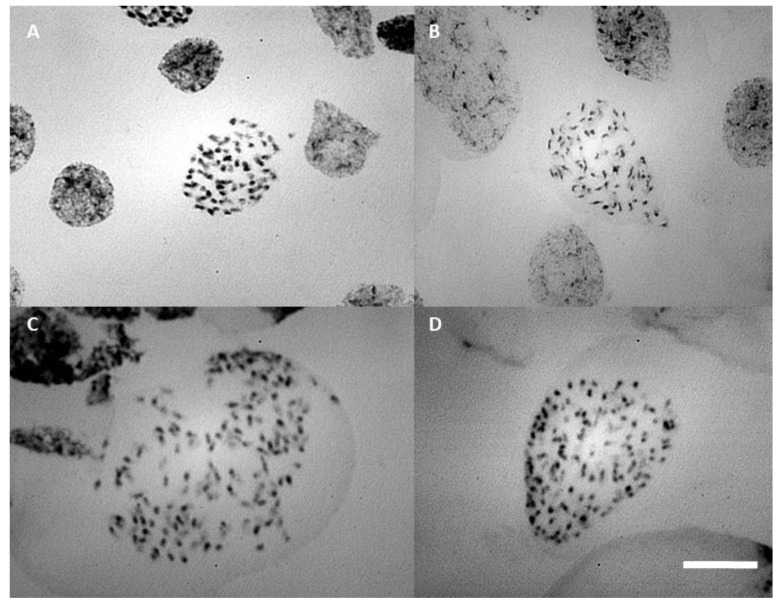
Prometaphase chromosome: (**A**) progeny of *Blc.* Haw Yuan Beauty × *Blc.* Goldenzelle “LC” without colchicine treatment; (**B**) *Blc*. Haw Yuan Beauty; (**C**,**D**) progeny of *Blc.* Haw Yuan Beauty × *Blc*. Goldenzelle “LC” treated with colchicine. Scale bar = 10 μm.

**Figure 7 plants-12-00281-f007:**
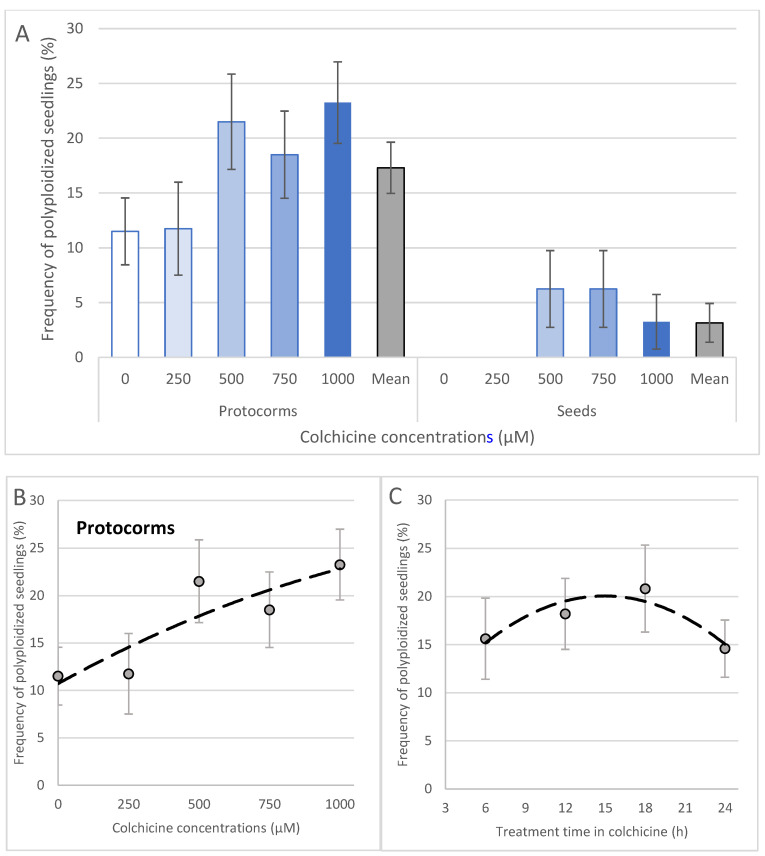
Polyploidy frequency in seedlings of *Blc*. hybrid. The type of the explant and concentration of colchicine in the frequency of polyploidization (**A**) and correlation with concentrations (**B**) and treatment time (**C**) in protocorms treated with colchicine. The Student’s *t*-test showed significant values of coefficient of correlation (r) (0.05 *) for protocorms (treatment time and colchicine concentration) and for seeds (only for colchicine concentration). Equations and r values: (**B**) y = −4 × 10^−6^x^2^ + 0.0164x + 10.714, *r* = 0.882 *; (**C**) y = −0.0611x^2^ + 1.8267x + 6.4, *r* = 0.913 *.

**Figure 8 plants-12-00281-f008:**
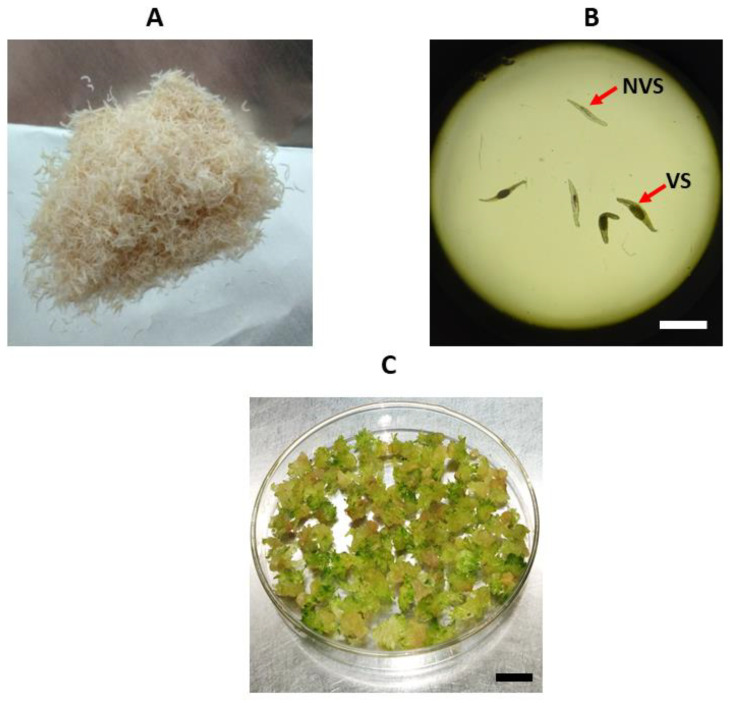
Types of explants used for polyploidy induction: (**A**) macroscopic view of the seeds; (**B**) microscopic view (×40) of viable seeds (VS) containing embryos and non-viable seeds (NVS); (**C**) protocorms used in colchicine treatments. Scale bar = 1mm (**B**) and Scale bar = 1 cm (**C**).

**Figure 9 plants-12-00281-f009:**
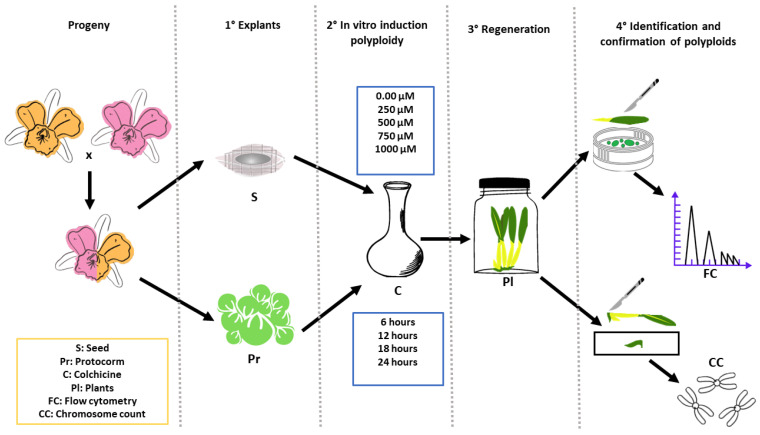
Graphical abstract of the main procedures, treatments and analysis of *Blc*. progeny treated with colchicine aiming polyploidization.

**Table 1 plants-12-00281-t001:** Germination percentage (GP), percentages of seeds developed into protocorms (PrD) and into seedlings (SeD) and total fresh weight (TFW) of *Blc.* Haw Yuan Beauty x *Blc.* Goldenzelle “LC” seed progeny exposed to different times and concentrations of colchicine.

	**Seeds**			
**Colchicine Treatment Time (h)**	**GP (%)**	**PrD (%)**	**SeD (%)**	**TFW (g)**
6	44.86 a	54.99 c	45.01 a	13.62 a
12	31.53 c	62.51 ab	39.15 ab	12.40 a
18	29.94 c	58.25 bc	41.75 a	12.26 a
24	36.96 b	68.36 a	31.64 b	13.42 a
**Colchicine concentrations (μM)**				
0.00	49.42 a	62.88 a	37.10 b	14.52 a
250	36.21 b	60.38 ab	39.62 ab	12.76 ab
500	34.86 b	67.21 a	34.86 b	13.08 ab
750	30.64 c	61.75 a	38.25 b	12.50 ab
1000	27.97 d	52.91 b	47.09 a	11.77 b
F Colchicine treatment time (h)	59.15 **	12.91 **	10.06 **	2.49 ns
F Colchicine concentrations (%)	69.89 **	7.36 **	4.78 **	3.85 **
F Interaction	5.13 **	4.1 **	3.10 **	1.64 ns
CV (%)	3.28	24.19	4.05	5.96

Mean values followed by different letters, in the same column, are significantly different by Tukey’s test at 1% probability. ** significant at 1% probability (*p* < 0.01).

**Table 2 plants-12-00281-t002:** Percentages of survival (PS), PLBs proliferation (PLBP), regeneration into plantlets (RlP) and total fresh weight (TFW) of *Blc.* Haw Yuan Beauty × *Blc.* Goldenzelle “LC” progeny protocorms exposed to different times and concentrations of colchicine.

	**Protocorm**			
**Colchicine Treatment Time (h)**	**PS (%)**	**PLBP (%)**	**RIP (%)**	**TFW (g)**
6	64.67 a	59.38 a	40.62 b	10.07 ab
12	59.50 a	36.94 b	63.06 a	11.70 ab
18	63.34 a	46.91 ab	53.09 ab	8.92 b
24	61.67 a	40.92 b	59.08 a	12.30 a
**Colchicine concentrations (μM)**				
0.00	72.92 a	44.19 a	55.80 a	12.75 a
250	71.25 a	47.21 a	52.79 a	11.01 ab
500	66.67 ab	43.47 a	56.53 a	11.23 ab
750	51.04 bc	46.33 a	53.67 a	10.07 ab
1000	49.58 c	48.97 a	51.03 a	8.66 b
F treatment time	0.49 ns	5.0 **	5.25 **	4.07 *
F concentrations	7.66 **	0.21 ns	0.22 ns	2.30 *
F Interaction	1.92 *	0.67 ns	0.70 ns	0.96 ns
CV (%)	15.97	41.11	33.36	22.43

Mean values followed by different letters, in the same column, are significantly different by Tukey’s test at 1% and 5% probability levels. ** significant at 1% probability (*p* < 0.01); * significant at 5% probability (*p* < 0.05).

## Data Availability

Data associated with this study may also be found at https://repositorio.ufscar.br/bitstream/handle/ufscar/16352/Escrita.DissertacaoFINAL04-07-22.pdf?sequence=1&isAllowed=y (accessed on 3 January 2023).

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
