# Peer review of "In Vitro Polyploidization of Brassolaeliocattleya Hybrid Orchid"

_plants, 2023, doi:10.3390/plants12020281_

Round 1

Reviewer 1 Report

The authors of the manuscript “Efficient in vitro polyploidization of Brassolaeliocattleya orchid in response to the type of explant and colchicine” established the protocol for the generation of Brassolaeliocattleya polyploids. The authors treated seeds and protocorms with 0.05-0.08% colchicine for 6-24 h, and the polyploids were identified by flow cytometric analysis and chromosome counting. The results are encouraging. However, the statistical analysis is missing (mean separation).

Specific comments:

L19: Orchidaceae (nonitalic). Please correct it throughout the manuscript.

L30: in vitro (nonitalic). Please correct it throughout the manuscript.

L120: 0.06%, or 0.08%, 18, or 24

L121-122: Please provide the micrographs of seeds and protocorms for better understanding.

L222: Please define the abbreviation at first use ‘PLBs’ (protocorms like bodies or secondary protocorms).

L252: Please indicate the seedlings age.

L330: Mondin and Aguiar -Perecin [29] instead of Mondin and Aguiar -Perecin (2009).

L345: ‘seeds containing embryos’ Add a method for identifying embryos.

L347: Please provide the data on the effects of colchicine concentration and exposure time (supplementary Tables showing 0.02% at 0, 6, 12, 18, and 24 h, 0.04% at 0, 6, 12, 18, and 24 h, 0.06% at 0, 6, 12, 18, and 24 h, 0.08% at 0, 6, 12, 18, and 24 h seeds and protocorms) for better understanding.

L381: The error bars represent? Please indicate (Figure 1).

L475: The error bars represent? Please indicate (Figure 2).

L557: The error bars represent? Please indicate (Figure 3).

L562: The error bars represent? Please indicate (Figure 4).

L677: Please improve the quality of figure 5 and correct the legends.

L681: Please introduce scale bars (Figure 6). Also, Figures 8 and 11.

L1188-1190: Please indicate the shape, dimensions, and coloration. It is tough to find out the morphological differences in Figure 11.

Author Response

Reviewer 1

The authors of the manuscript “Efficient in vitro polyploidization of Brassolaeliocattleya orchid in response to the type of explant and colchicine” established the protocol for the generation of Brassolaeliocattleya polyploids. The authors treated seeds and protocorms with 0.05-0.08% colchicine for 6-24 h, and the polyploids were identified by flow cytometric analysis and chromosome counting. The results are encouraging. However, the statistical analysis is missing (mean separation).

Response: Thanks for your valuable review. The previous figures present the standard error in each point of visualization and used to differentiate the treatments. However, follow our suggestions we provide in this new version the tables containing the values and statistical mean analysis.

L19: Orchidaceae (nonitalic). Please correct it throughout the manuscript.

Response: corrected.

L30: in vitro (nonitalic). Please correct it throughout the manuscript.

Response: corrected.

L120: 0.06%, or 0.08%, 18, or 24

Response: the text was corrected.

L121-122: Please provide the micrographs of seeds and protocorms for better understanding.

Response: A figure (Figure 8) was provided showing the explants used for the induction of polyploids, including the micrography of seeds.

L222: Please define the abbreviation at first use ‘PLBs’ (protocorms like bodies or secondary protocorms).

Response: the word “PLBs” was defined as protocorms like bodies.

L252: Please indicate the seedlings age.

Response: the age of the plants was indicated. See ‘Eight seedlings (3-4 months age) from each treatment of seeds and 13 seedlings (7-8 months age) from each treatment of the protocorms in MS culture medium were analyzed.’

L330: Mondin and Aguiar -Perecin [29] instead of Mondin and Aguiar -Perecin (2009).

Response: the text was corrected.

L345: ‘seeds containing embryos’ Add a method for identifying embryos.

Response: the method used to identify the embryos of the seeds was added: Counting method with the help of an optical microscope with a 4X objective.

L347: Please provide the data on the effects of colchicine concentration and exposure time (supplementary Tables showing 0.02% at 0, 6, 12, 18, and 24 h, 0.04% at 0, 6, 12, 18, and 24 h, 0.06% at 0, 6, 12, 18, and 24 h, 0.08% at 0, 6, 12, 18, and 24 h seeds and protocorms) for better understanding.

Response: two tables were provided (Table 1 and 2) showing the variables evaluated for each type of explants, including statistical analysis containing mean test comparison.

L381: The error bars represent? Please indicate (Figure 1).

Response: The error of the bars represents the standard error of the mean for each variable.

L475: The error bars represent? Please indicate (Figure 2).

Response: The error of the bars represents the standard error of the mean for that variable.

L557: The error bars represent? Please indicate (Figure 3).

Response: The error of the bars represents the standard error of the mean for that variable.

L562: The error bars represent? Please indicate (Figure 4).

Response: The error of the bars represents the standard error of the mean for that variable.

L677: Please improve the quality of figure 5 and correct the legends.

Response: the quality of figure 5 was improved and the legends were corrected

L681: Please introduce scale bars (Figure 6). Also, Figures 8 and 11.

Response: a scale bar was introduced for figures 6, 8 and 11.

L1188-1190: Please indicate the shape, dimensions, and coloration. It is tough to find out the morphological differences in Figure 11.

Response: The follow sentence was introduced in the text. The figure was also improved. Polyploid plants had oblong-shaped leaves with greater thickness, width, and intensity of green coloration (Figure 5A), unlike non-polyploid plants that had lanceolate-shaped leaves with less width, thickness, and light green coloration (Figure 5B).

Reviewer 2 Report

The manuscript 'Efficient in vitro polyploidization of Brassolaeliocattleya orchid in response to the type of explant and colchicine' refers to the significant commercial use of scientific achievements in terms of ornamental species.

Formatting of the text is confusing. Line numbering, results in Discussion, etc.

Unfortunately, the text is poorly written and hard to understand. Moreover, there is chaos in the manuscript - why results are presented and described in 'Discussion' section? And I don't understand the main purpose of this work; efficient polyploidization? If so, why authors admitted in lines 914-915 that longer exposure to colchicine might improve polyploidization, but in this work only shorter time was analyzed.

I wanted to put my comments in the PDF file to highlights unintelligible fragments, but unfortunatelly all manuscript needs to be improved and rewrite.

I suggest to remove Fig.3B and Fig.4B, improve quality of Fig.5. Something is wrong with Fig.7B and D: Y-axis 'Average'? On e.g., Fig.3 Fresh weight in %?

For better understand all aspects of this work, I recommend to prepare some scheme of all steps, because it is hard to follow - what is the concentration of colchicine in Fig. 2 and 4. 

All together, in this form I recommend to reject this paper.

Author Response

Reviewer 2

Dear Reviewer 2, we would like to thanks you for changes required and contributions to our manuscript. We hope we attend your queries in this new reviewed version. Best regards, Jean Cardoso.

Formatting of the text is confusing. Line numbering, results in Discussion, etc.

Response: We improved the formatting for avoid confusion. The line numbering was deleted. We separated and rewriting discussion section, as suggested. However, the discussion was based on our results, and sometimes, we included this type of information for effort the discussion.

Unfortunately, the text is poorly written and hard to understand. Moreover, there is chaos in the manuscript - why results are presented and described in 'Discussion' section?

Response: We improved the text for better meaning and to avoid confusion.

And I don't understand the main purpose of this work; efficient polyploidization? If so, why authors admitted in lines 914-915 that longer exposure to colchicine might improve polyploidization, but in this work only shorter time was analyzed.

Response: The main purpose of this work is to obtain a protocol for polyplodization in Blc. orchids, with no knowledge in the present published data, and also study the effects of the type of explants submitted to different concentrations and time of exposure of colchicine. The affirmative in lines 914-915 which you appointed is referent to seeds (one type of explants), that in fact, other authors obtained better results with Calanthe genus (Orchidaceae) using higher times of exposition. However, see that our results obtained with Blc. progeny, obtained one of the best rates of polyploidization when protocorms were used as explants (near 50%), compared with other authors. However, we understood the reviewer appointment, and deleted the word ‘Efficient’ from the title and objectives to improve the meaning. Also, the aim of the study was improved and rewritten for better meaning.

I wanted to put my comments in the PDF file to highlights unintelligible fragments, but unfortunatelly all manuscript needs to be improved and rewrite.

Resposne: The text was completely revised by authors and editing service company.

I suggest to remove Fig.3B and Fig.4B, improve quality of Fig.5. Something is wrong with Fig.7B and D: Y-axis 'Average'? On e.g., Fig.3 Fresh weight in %?

Response: We replaced Figures by two tables containing mean test analysis (Table 1 and 2). These tables summarize all the variables analyzed according to each explant treated with colchicine. The quality of figure 5 was improved. Figure 7 has been modified and contains a scatterplot. The "fresh weight" titer was corrected for percentage of PLBs proliferation and the regeneration into plantlets.

For better understand all aspects of this work, I recommend to prepare some scheme of all steps, because it is hard to follow - what is the concentration of colchicine in Fig. 2 and 4. 

Response:

  Response: For a better understanding of the entire methodology used, a graphical abstract will be included showing all the steps that were used to obtain the Blc. polyploids.

Reviewer 3 Report

The paper required to be reviewed by a native English speaker. There are several important grammatical errors throughout the manuscript.

The manuscript does not follow the journal format. It has double numbering on the lines. The items in Materials and Methods do not correspond to the consecutive number.

The methodology does not indicate the companies, places of origin, and catalog numbers for all the reagents, only for some.

Some specific comments are listed below, and the line number in which they are found.

Lines 13 – 18, 71 - 76, 88 – 95, 101, 120 – 129, 136 - 140, 162 – 165, 298, 307, etc. Table 1, Table 2, Table 3. The concentration of growth regulators must be expressed in molarity.

Lines 114, 222. The first time an abbreviation is used, it must be written in full. For example, MS should be Murashige and Skoog (MS).PLBs (222).

Line 119. Dimethysulfoxide must be written dimethylsulfoxide.

Line 330. The number for the reference for Mondin and Aguiar-Perecin (2009) is missing.

Line 341. The P for statistical presentation must be P.

Line 667. The quality of the figures 5 & 10 must be improved.

Line 557, 562. Why use two graphs to indicate the same phenomenon? Figures 3A and 3B show the same biological phenomenon from two viewpoints. Only one should be used.

Line 786. Discussion. The authors include figures 7 to 11 in the discussion when, in fact, they are part of the results. Discussion only occurs at the end of each paragraph in which these figures are presented. This section should be rewritten, separating the results of the discussion. It is not that it cannot be done that way. This is the format of the magazine. If the authors wish to make a single section, they should consult with the journal editors. Nevertheless, in any case, all the results should be included in the same section as the discussion and not just some, as is the case in this paper.

The manuscript requires a revision of the English. As an example, this reviewer has edited the abstract.

Abstract

The Cattleya (Orchidaceae – Laeliinae subtribe) intergeneric hybrids, such as Brassolaeliocattleya (Blc.), have great ornamental value among orchids,  due to their compact size, large and high color diversity of flowers. The main limitations to expanding the Cattleya hybrids market have been the long juvenile period (5-10 years) and the short shelf-life of their flowers (15-20 days), which could be solved by the use of polyploid plants the use of polyploid plants could solve. To obtain an efficient polyploidization protocol, this study aimed to test different types of explants (seeds and protocorms), concentrations, and exposure times to colchicine. In addition, the in vitro environment provides more efficient polyploidization methods, than in vivo ones. The aim of this study was to test different types of explants (seeds and protocorms), concentrations and exposure times to colchicine to obtain an efficient polyploidization protocol. At the same time, ploidy levels, and polyploidization due to hybridization, as well asand the consequences of polyploidization are still limitedly reported in the Laellineae subtribe of orchids. The present study obtained more than one protocol aiming at the polyploidization of Blc. orchids. The highest percentage of polyploid plants was regenerated from protocorms treated with colchicine, instead of seeds. Also, the natural in vitro polyploidization was reported using protocorms. Also, it was possible to compare the ploidy level of parents and progenies used in this study, and polyploid plants treated with colchicine, using flow cytometry, as well as estimate the number of chromosomes for each one.

Author Response

Reviewer 3

Dear Reviewer 3, we would like to thanks you for changes required and contributions to our manuscript. We hope we attend your queries in this new reviewed version. Best regards, Jean Cardoso

The paper required to be reviewed by a native English speaker. There are several important grammatical errors throughout the manuscript.

Response: The paper was reviewed/edited by a professional English service to provide these corrections.

The manuscript does not follow the journal format. It has double numbering on the lines. The items in Materials and Methods do not correspond to the consecutive number.

Response: We follow the journal format, eliminating these appointments observed by reviewers.

The methodology does not indicate the companies, places of origin, and catalog numbers for all the reagents, only for some.

Response: We was added this information for most of the reagents used in our study.

Some specific comments are listed below, and the line number in which they are found.

Lines 13 – 18, 71 - 76, 88 – 95, 101, 120 – 129, 136 - 140, 162 – 165, 298, 307, etc. Table 1, Table 2, Table 3. The concentration of growth regulators must be expressed in molarity.

Response: As recommended, we change this information to micromolarity: 250 μM, 500 μM, 750 μM or 1000 μM.

Lines 114, 222. The first time an abbreviation is used, it must be written in full. For example, MS should be Murashige and Skoog (MS).PLBs (222).

Response: ok

Line 119. Dimethysulfoxide must be written dimethylsulfoxide.

Response: ok

Line 330. The number for the reference for Mondin and Aguiar-Perecin (2009) is missing.

Response: the text was corrected for Mondin and Aguiar-Perecin [51]

Line 341. The P for statistical presentation must be P.

Response: ok

Line 667. The quality of the figures 5 & 10 must be improved.

Response: The quality of figures 5 and 10 were improved, as suggested.

Line 557, 562. Why use two graphs to indicate the same phenomenon? Figures 3A and 3B show the same biological phenomenon from two viewpoints. Only one should be used.

Response: We change these figures to avoid the demonstration of the same biological phenomena.

Line 786. Discussion. The authors include figures 7 to 11 in the discussion when, in fact, they are part of the results. Discussion only occurs at the end of each paragraph in which these figures are presented. This section should be rewritten, separating the results of the discussion. It is not that it cannot be done that way. This is the format of the magazine. If the authors wish to make a single section, they should consult with the journal editors. Nevertheless, in any case, all the results should be included in the same section as the discussion and not just some, as is the case in this paper.

Response: We manage to place and rewrite the text according to the journal template and we separate the results and the discussion to avoid confusion and make the reading of the text more fluid. In the discussion section the figures were replace to results section. Also, the discussion section are extensive rewritten and edited to provide better quality text.

Round 2

Reviewer 1 Report

The authors have addressed all the comments. 

Author Response

Thanks for your precious revision and for the improvement of our manuscript

Reviewer 2 Report

I suggest authors check the style again, e.g. (paragraph 2.2) 'the seedlings obtained' - should be 'the obtained seedling', etc.

After revision paper is more readable, thus I recommend accepting this manuscript to publish in Plants.

Author Response

The style was reviewed and the changes added to the text. Thanks again for your value commentaries for improving our manuscript.

Reviewer 3 Report

None

Author Response

Thanks for your valuable commentaries that improving the quality of our manuscript